



# A large-area blackbody for inflight calibration of an infrared interferometer deployed on board a long-duration balloon for stratospheric research

Friedhelm Olschewski[1], Christian Monte[2], Albert Adibekyan[2], Max Reiniger[2], Berndt Gutschwager[2], Joerg Hollandt[2], and Ralf Koppmann[1]

[1]Institute for Atmospheric and Environmental Research at the University of Wuppertal, 42097 Wuppertal, Germany
[2]Physikalisch-Technische Bundesanstalt, 10587 Berlin, Germany

*Correspondence to:* F. Olschewski (olsch@uni-wuppertal.de)

**Abstract.** The deployment of the imaging Fourier Transform Spectrometer GLORIA (Gimballed Limb Observer for Radiance Imaging of the Atmosphere) on board a long-duration balloon for stratospheric research requires a blackbody for inflight calibration in order to provide traceability to the International Temperature Scale (ITS-90) to ensure comparability with the results of other experiments and over time. GLORIA, which has been deployed onboard various research aircraft such as the

Russian M55 Geophysica or the German HALO in the past, shall also be used for detailed atmospheric measurements in the stratosphere up to 40 km altitude. The instrument uses a two-dimensional detector array and an imaging optics with a large aperture diameter of 36 mm and an opening angle of $4.07° × 4.07°$ for infrared limb observations. To overfill the field-of-view (FOV) of the instrument, a large-area blackbody radiation sources ($125\,\text{mm} × 125\,\text{mm}$) is required for inflight calibration.

In order to meet the requirements regarding the scientific goals of the GLORIA missions, the radiance temperature of the black-

body calibration source has to be determined to better than 100 mK and the spatial uniformity shall be better than 100 mK. Since electrical resources onboard a stratospheric balloon are very limited, the latent heat of the phase change of a eutectic material is utilized for temperature stabilization of the calibration source, such that the blackbody has a constant temperature of about -32 °C corresponding to a typical temperature observed in the stratosphere.

The Institute for Atmospheric and Environmental Research at the University of Wuppertal designed and manufactured a proto-

type of the large-area blackbody for inflight calibration of an infrared interferometer deployed onboard a long-duration balloon for stratospheric research. This newly developed calibration source was tested under lab conditions as well as in a climatic and environmental test chamber in order to verify its performance especially under flight conditions. At PTB (Physikalisch-Technische Bundesanstalt), the German national metrology institute the spectral and spatial radiance distribution of the blackbody was determined and traceability to the International Temperature Scale (ITS-90) has been assured. In this paper the design

and performance of the Balloon-borne BlackBody (BBB) is presented.



## 1 Introduction

The Gimballed Limb Observer for Radiance Imaging of the Atmosphere (GLORIA) is an imaging Fourier Transform Spectrometer (FTS) developed for trace gas measurements in the atmosphere. GLORIA utilizes a two-dimensional MCT detector array with a large aperture optical system for detailed infrared limb observations (Riese et al., 2014; Friedl-Vallon et al.,
2006). In the past, GLORIA participated in various international research campaigns (Friedl-Vallon, 2016) (e.g. TACTS 2012 (Kaufmann et al., 2015), POLSTRACC 2015 (Krisch et al., 2017), STRATO-CLIM 2016 (Johansson et al., 2017), WISE 2017) deployed on board the research aircraft Geophysica and HALO. It is planned to install GLORIA on board a stratospheric balloon in order to perform long-term measurements at altitudes up to $40\,\mathrm{km}$ for several weeks. Since the environmental conditions at the flight altitude ($35 - 38\,\mathrm{km}$) are very different from those on board a research aircraft, a new concept for the inflight cal-
ibration system had to be developed. Especially the limitation of electrical power was an important factor in the design of the new large-area blackbody calibration source.

## 2 The GLORIA instrument

GLORIA is a joint project of Research Center Juelich and Karlsruhe Institute of Technology, Germany. GLORIA is designed as imaging FTS operating in the thermal infrared spectral region from $770\,\mathrm{cm}^{-1}$ to $1400\,\mathrm{cm}^{-1}$ (Friedl-Vallon et al., 2014).
The spectral resolution is adjusted to two different measuring modes: $1.25\,\mathrm{cm}^{-1}$ for the dynamics mode and $0.1\,\mathrm{cm}^{-1}$ for the chemistry mode respectively. A two-lens aspherical telescope with an aperture of approximately $36\,\mathrm{mm}$ and a vertical and horizontal field of view (FOV) of $4.07° \times 4.07°$ is used to image the atmosphere. The radiation coming from the atmosphere is directly projected onto a large two-dimensional photovoltaic MCT detector array mounted in a dewar with an integrated Stirling cooler. GLORIA makes novel information on small-scale atmospheric dynamics available, e.g. STE, the Stratosphere-
Troposphere Exchange and other important phenomena (Riese et al., 2014; Ungermann et al., 2011, 2015).
In order to study long-term phenomena in the stratosphere for up to two weeks, GLORIA shall be installed into a gondola of a stratospheric balloon. A schematic of the balloon instrument is shown in Figure 1. The GLORIA balloon instrument utilizes a rotatable mirror for line-of-sight stabilization during measurement. This mirror is also used to adjust the line-of-sight to the blackbody calibration source and to view deep space.

## 25  3  Calibration concept and requirements

In order to retrieve temperature and trace gas concentrations in the atmosphere, the measured detector signals need to be converted into atmospheric infrared radiance spectra with very small uncertainties. Assuming that the measurement system has a linear response, the required high accuracy can be achieved by a two point calibration in the range of the observed atmospheric radiance (Kleinert et al., 2014). Deep space as an ideal blackbody with a very uniform temperature of $2.7\,\mathrm{K}$ is a perfect cali-
bration source for the detector. As second calibration source a large-area blackbody at a temperature of about $240\,\mathrm{K}$ with high temperature homogeneity of better than $0.1\,\mathrm{K}$ shall be used.



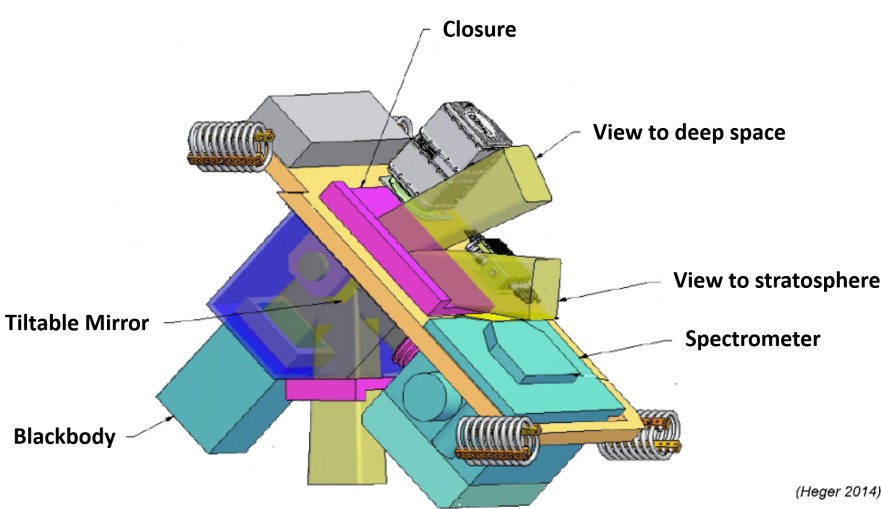

**Figure 1.** Schematic of the GLORIA balloon instrument

Since the uncertainties in spectral radiance of the blackbody calibration source influences the uncertainties in the measured atmospheric radiances, radiometric calibration of the instrument with an uncertainty of less than 1 % is necessary (Olschewski et al., 2012). This requires a very high emissivity of greater than 0.99 and very precise temperature measurement with a temperature uncertainty of less than 0.1 K. For the technical realization of the radiometric accuracy, the large-area blackbody calibration
5   source has to fulfill the requirements listed in Table 1.

## 4   Design of the GLORIA Balloon Blackbody

The GLORIA inflight calibration system uses a high-precision blackbody radiation source, which is operated at a temperature of about 240 K. The optical surface of this Balloon BlackBody (BBB) consists of a wire-eroded aluminum plate with an array of 225 small pyramids, which are varnished with NEXTEL-Velvet Coating (see Figure 2). Integrated in a housing with outer
10   dimensions of $140 \, \text{mm} \times 140 \, \text{mm} \times 200 \, \text{mm}$, the optical surface has an effective emissivity of $\epsilon > 0.997$. In order to verify the thermal uniformity, ten thermally cycled and calibrated platinum resistance thermometers (PRTs) are installed in the aluminum plate.



**Table 1.** Requirements for the large-area blackbody calibration source

| | |
|---|---|
| Optical surface | 125 mm × 125 mm |
| Temperature range | -30 °C to -35 °C |
| Temperature uncertainty | < 0.1 K |
| Effective emissivity (7 µm - 13 µm) | > 0.99 |
| Spatial temperature uniformity | < 0.15 K |
| Short-term temperature stability | < 25 mK/min |

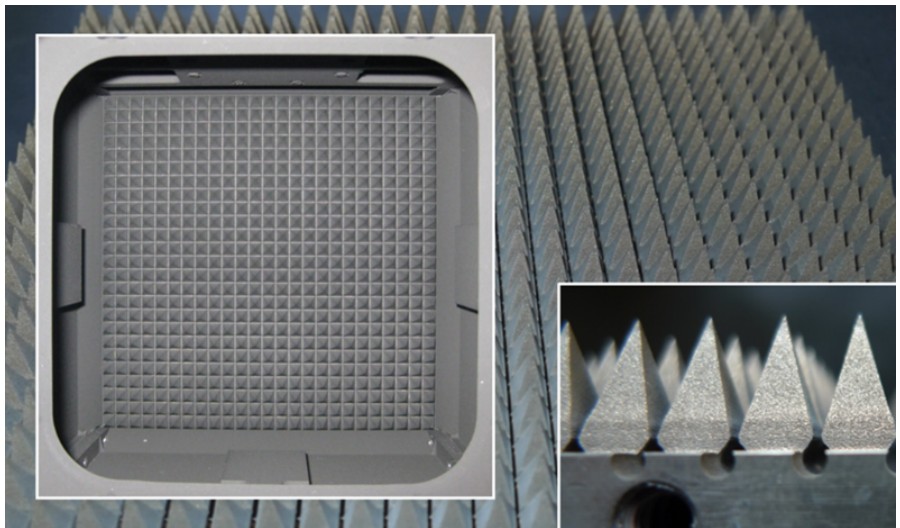

**Figure 2.** Optical surface of large-area blackbody with wire-eroded pyramid array

For thermal decoupling of the blackbody calibration source from the GLORIA balloon instrument, Glass-Fiber Reinforced Plastic (GFRP) parts are used. In order to reduce the adverse influence of the thermal environment, the BBB is covered with polystyrene foam sheets (see Figure 3).

For temperature stabilization of the airborne GLORIA Blackbodies (GBBs) thermo-electric coolers (TECs) are used (cf. Olschewski et al., 2013). This concept is not feasible for the balloon-borne instrument because the power consumption is too


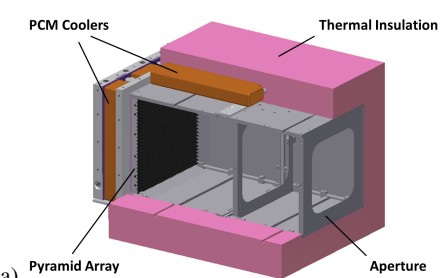
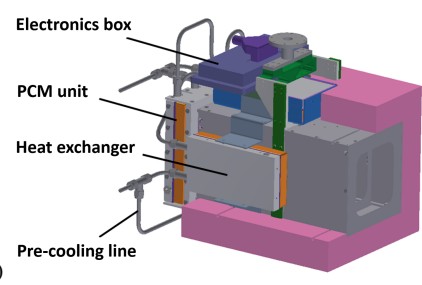

**Figure 3.** Design of Balloon Blackbody (a) with electronics and heat exchangers for precooling (b)

high and there is not sufficient air for cooling the heat exchangers at flight altitude. Therefore, a different design concept is realized. The phase change of a eutectic material is used to stabilize the temperature of the optical surface at about -32 °C. Six commercial cooling pads from Va-Q-tec are used for thermal stabilization (https://www.va-q-tec.com/de/produkte/kaelte-und-waermespeicher/va-q-accu-32g.html). Table 2 gives the properties of the PCM (Phase Change Material) cooling pads. Two

5 cooling pads are used for temperature control of the optical surface while one each will control the temperature of the four walls of the housing.

**Table 2.** Properties of the PCM cooling pads "va-Q-accu -32G"

| | |
|---|---|
| freezing / melting point | -32 °C |
| heat capacity (solid) | $2.95\,\mathrm{kJ}/(\mathrm{kg}*\mathrm{K})$ |
| latent heat | $243\,\mathrm{kJ/kg}$ |
| dimensions | $165\,\mathrm{mm}\times88\,\mathrm{mm}\times20\,\mathrm{mm}$ |
| PCM mass | $0.28\,\mathrm{kg}$ |

In order to get the PCM into the solid state, pre-cooling with a chiller or with liquid nitrogen is needed. Therefore, the PCM cooling pads are equipped with heat exchangers made of aluminum shown in Figure 3b.

## 5  Lab tests and test in a thermal vacuum chamber

10 In order to study the thermal behavior of the BBB, thermal tests were performed in the lab and in the thermal vacuum chamber of Research Centre Juelich. It was the goal of these tests to measure the progression of the phase change and to estimate the possible operating time on a balloon mission in the stratosphere. Figure 4a shows the thermal behavior during a lab test while





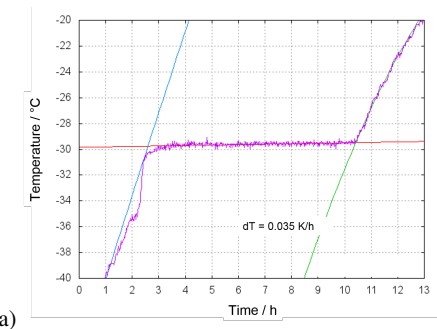
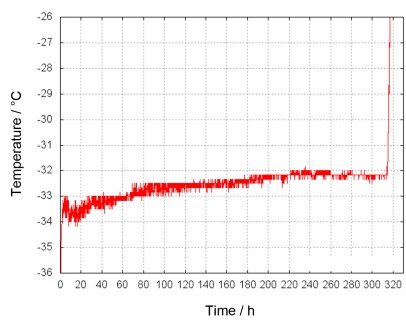

a)
b)

**Figure 4.** Lab test of the melting plateau at ambient pressure of about 1013 hPa and ambient temperature of about 23 °C (a) and thermal-vacuum test of the BBB at ambient pressure of 10 hPa and ambient temperature of about -22 °C (b)

the aperture was closed with a polystyrene foam sheet. The pre-cooling of the PCM cooling pads was achieved by storing the complete device in a cooling box for 12 hours. The progression of the phase change lasted seven hours with a very small temperature change of 35 mK/h. In the thermal vacuum chamber the most realistic condition of a balloon flight was simulated. The chamber pressure was set to 10 hPa and the temperature to -22 °C, respectively. The complete temperature evolution over

5    time is shown in Figure 4b. Unfortunately the test had to be discontinued after 320 hours for organizational reasons. At the beginning of the test, the mean BBB temperature was about -33.5 °C. The phase change started nearly a week later and was still ongoing when the test ended. Since the temperature trend in the solid state was very small, calibrating the GLORIA instrument before the onset of the phase change during a balloon flight will also be possible.

## 6   Radiometric Characterisation

10    The radiometric characterisation of the large-area blackbody was performed inside the Reduced Background Calibration Facility (RBCF) (Hollandt et al., 2003/2004; Monte et al., 2009) of the Physikalisch-Technische Bundesanstalt (PTB) as shown in Figure 5. The pressure inside the RBCF was set to 10 hPa corresponding to the ambient pressure expected in the stratosphere. The heat exchangers of the blackbody calibration source were connected to an external chiller. By adjusting the temperature of the coolant, the PCM elements were taken to the solid state first and then the chiller thermostat was set slightly above the

15    melting point. Due to its large optical surface, the blackbody faces a thermally non-uniform environment inside the RBCF with temperatures in the range between -120 °C and 23 °C. The radiation exchange with the environment through the aperture is neither identical to the operating conditions in the stratosphere nor to the measurement setup in the thermal-vacuum chamber. In spite of this drawback, a phase change time (melting-plateau) of more than eight hours could be reached. During this phase change period the radiance temperature and its lateral distribution was measured.





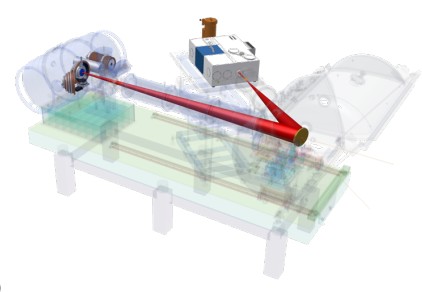

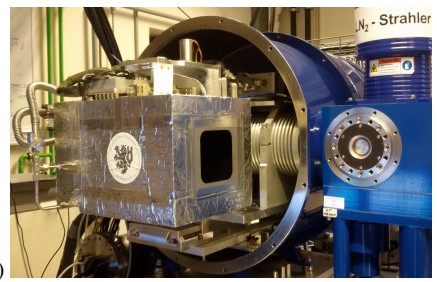

a)                                                                                        b)

**Figure 5.** (a) The Reduced Background Calibration Facility (RBCF) at PTB and (b) The large-area balloon blackbody inside the source chamber of the RBCF

The change of radiance temperature and of contact temperature over time is shown in Figure 6 together with the set temperature of the external chiller for thermalizing the heat exchangers. The collinearity of radiance temperature and contact temperature is obvious.

Figure 7 demonstrates the very high spatial uniformity of the radiance temperature of better than 100 mK (peak-to-peak) over the used area of the aperture of the blackbody. The sampling of the radiating surface of the blackbody was performed by the calibrated broad band radiation thermometer VIRST (8 - 14 µm) (Gutschwager et al., 2008). These results show the suitability of the large-area blackbody BBB for the radiometric traceability of balloon-borne imaging spectrometers as GLORIA.

The blackbody should not be considered as reference blackbody operating at a fixed temperature given by the melting plateau of the PCM material. Rather, the PCM material should be considered as a reservoir for latent heat only, enabling the operation of the blackbody with very low energy consumption. By using well characterized PTR sensors in the backplane which are absolutely calibrated with low uncertainties, the momentary temperature of the backplane can be determined (cf. Figure 6) and the corresponding radiance temperature is given as well by the calibration in terms of radiance temperature. Additional radiometric characterizations of the lateral distribution of the radiance temperature in the liquid and frozen state revealed also a very good uniformity. This would permit the use of the blackbody even under these conditions.

## 7   Summary and conclusions

The Institute for Atmospheric and Environmental Research at the University of Wuppertal developed a large-area calibration source based on phase change material for deployment on board stratospheric balloons. The newly designed blackbody can be used for precise inflight calibration of hyperspectral cameras which are assigned for remote sensing of the atmosphere. The use of phase change material enables a long-lasting temperature stability without any power consumption which is essential for long duration balloon flights. The radiometric characterisation at PTB showed that the requirement regarding the uncertainty





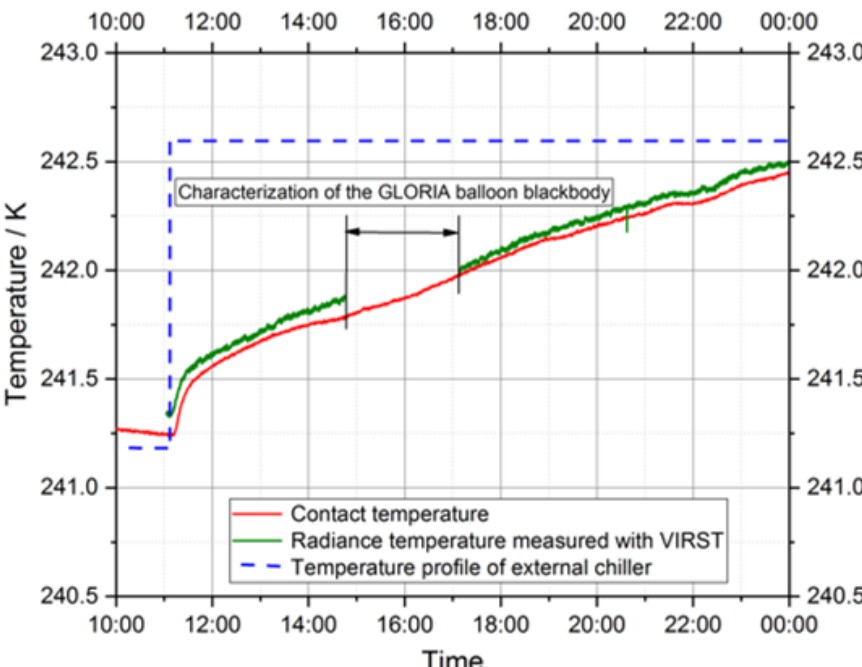

**Figure 6.** Change of radiance temperature and of contact temperature in backplane of the GLORIA balloon blackbody over time during phase change in PCM cooling pads.

in radiance temperature and its uniformity across the aperture below $100\,\mathrm{mK}$ can be reached. So atmospheric measurements employing this blackbody will become traceable to the International Temperature Scale (ITS-90) with low uncertainties.

*Acknowledgements.* Part of this work has been supported by the European Metrology Research Programme (EMRP) within the joint research project Metrology for Earth Observation and Climate (MetEOC2). The EMRP is jointly funded by the EMRP participating countries within EURAMET and the EU.



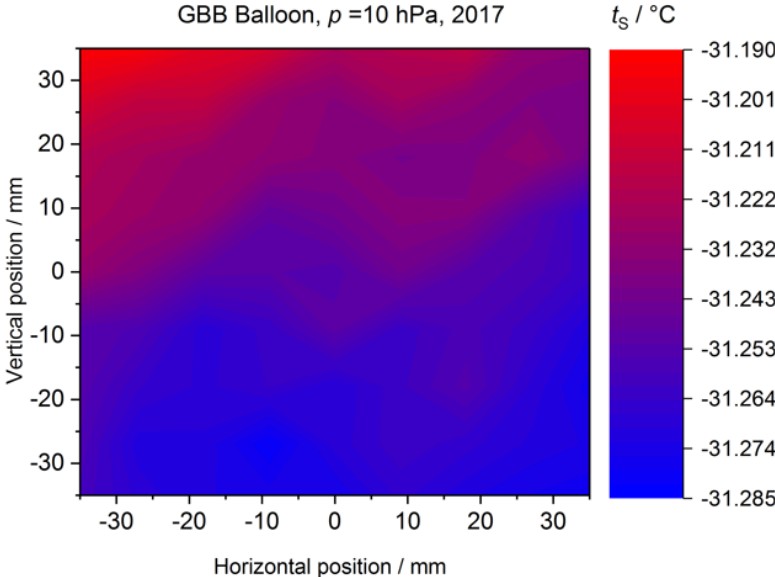

**Figure 7.** Lateral radiance temperature distribution of the large-area blackbody recorded during phase change featuring a non-uniformity of less than $100\,\mathrm{mK}$ (peak-to-peak).

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
