# Peer review of "A large-area blackbody for inflight calibration of an infrared interferometer deployed on board a long-duration balloon for stratospheric research"

_Atmospheric Measurement Techniques, 2017_

## Short Comment (SC1) · 8 May 2018

This paper presents a large-area blackbody designed for application on a stratospheric balloon. It is temperature stabilized using a phase change material. Some more explanations on the characterization measurements and their results would increase the value of the paper for the scientific community:

P.1 l.18: "... the spectral and spatial radience distribution of the blackbody was determined."

Can you give some results of the spectral distribution?

P.4 Tab.1: Here the required spatial temperature uniformity is < 0.15 K, while elsewhere in the text the requirement is given as < 0.1 K.

P.6 Fig.4: In the left plot, the plateau is at -30 °C, while it is at -32 °C in the right plot. What is the reason for this difference?

P.6, l.15/16: "Due to its large optical surface, the blackbody faces a thermally non-uniform environment inside the RBCF with temperatures in the range between -120 °C and 23 °C."

Can you explain where these large temperature differences come from? Generally some more words on the RBCF and the measurement setup would be helpful.

P.7, Fig.5a: Can you add some labels to the figure? Where is the source chamber, where is VIRST? And how is the temperature distribution inside the RCBF (see also comment above)?

P.7 l.2: "The collinearity of radiance temperature and contact temperature is obvious."

This is true, but there is a systematic difference of about 50 mK. Is this expected?

P.7 l.12: "the corresponding radiance temperature is given as well by the calibration in terms of radiance temperature"

What is meant by "the calibration in terms of radiance temperature"?

P.8 Fig.6: At the beginning of the measurement, the temperature of the external chiller is just around the freezing temperature of the PCM. Are you sure that the PCM is completely frozen?

As soon as the chiller temperature is raised, the temperature of the blackbody rises rather fast for about half an hour before reaching the melting plateau. I would expect the melting to start immediately. Can you coment on this behaviour?

The temperature gradient in this figure is about 75 mK/h which is more than twice as strong as in the lab (Fig. 4a) although the pressure is much lower and the ambient temperature is the same or lower. Is this expected?

P.9 Fig.7: The figure shows only 70 mm x 70 mm of the 125 mm x 125 mm optical surface. How is the temperature distribution outside the range shown?

Is the systematic gradient between upper and lower part of the optical surface expected? Is this gradient also reflected in the data of the 10 PRTs which are mentioned on p.3 l.11?

During the measurement of the lateral distribution (gap in Fig. 6), the overall blackbody temperature rises by about 100 to 150 mK. How is this temperature rise taken into account in order to make sure that the distribution shown in Fig. 7 is really a lateral effect and not interfering with an effect in time?

Typos:

P.1 l.18: A comma is missing after "institute"

P.3, l.1: "influence" without "s"

---

## Referee Comment (RC1) · Anonymous Referee #1 · 15 May 2018

This paper describes a newly developed large-area blackbody which is planned to be equipped on a stratospheric balloon instrument. Although the specification and lab-test of the blackbody is well described, the current manuscript looks like too much technical, and it seems the content could be of interest to a rather limited range of readers. To increase the value of the paper for a wider area of the scientific community, I would recommend to add some more descriptions with respect to the following points:

(1) Please give a brief background-history about what kind of calibrators have been used in past similar instruments (for example, MIPAS Balloon)

[Figure]

(2) Adding an introduction about the science objectives of balloon-borne GLORIA may help the readers to understand why careful radiance calibration is required. I understand that this paper does not aim at discussing scientific aspects of the GLORIA mission, but still I consider that any instrumental designs are optimized based on the scientific requirements and such information is important for readers. I could not follow well from where all the instrumental requirements' numbers (those summarized in Table 1) come from.

- Minor comments

p.1 L18: ...the German metrology institute"," the spectral and ...

p.4 Figure 2: if possible, please add the scale (length) on the close-up figure of pyramids.

p.7 Figure 5(a): please add some labels...it's hard to understand what exists in the experimental layout.

p.8 Figure 6: The vertical axis of figure may be expressed in celsius degree so that the comparison with Figures 4 and 7 become more easy.

---

## Referee Comment (RC2) · Anonymous Referee #2 · 13 Jun 2018

The manuscript describes the technical solution for calibrating a balloon-borne Fourier Transfom Spectrometer during its mission, including laboratory testing of the calibration hardware and procedure. The manuscript is well written, and the description meets scientific and academic standards, being understandable, repeatable, and with sufficient quantitative detail. I do agree with the very reasonable comments by the anonymous reviewer #1 and by Anne Kleinert. I recommend that the authors implement these in the final version and subsequently, I recommend the manuscript's acceptance.

I urge the authors to make a conscious decision on whether temperatures should be

given in K, in degrees Celsius, or both. They may consider using K throughout, which would presumably be the most systematic solution. Perhaps it is worthwhile to add the Celsius value in parentheses in certain cases. Celsius (Centigrade) are now used on p. 4 in Table 1, p. 5 line 2 and Table 2, p. 6 Figure 4 a), b) and figure caption, lines 4, 6, and 16, and Figure 7. The use of Celsius instead of K does not seem necessary in any of these cases, except perhaps in Tables 1 and 2, which give manufacturer specifications and may be given to whole Celsius values, but not the apparent 0.01 K precision that might result from calculation. I leave this to the authors to consider, but recommend that the choice is not made "by accident" in each individual case.

p. 2 l. 29/30. Given the field of view of about 4 degrees x 4 degrees, it seems impossible not to include stars, galaxies or other in the "deep space" calibration point, which might degrade the uniformity of the temperature field. Please add a comment of explanation.

typos:

p. 2 l. 14 consider "as an imaging FTS..."

p. 2 l. 16 consider "... the chemistry mode, respectively."

---

## Author Comment (AC2) · 29 Jun 2018

Thanks to Anonymous Referee #2 for the valuable comments. You will find our response below:

Comment:

I urge the authors to make a conscious decision on whether temperatures should be given in K, in degrees Celsius, or both. They may consider using K throughout, which would presumably be the most systematic solution. Perhaps it is worthwhile to add the

[Figure]

Celsius value in parentheses in certain cases. Celsius (Centigrade) are now used on p. 4 in Table 1, p. 5 line 2 and Table 2, p. 6 Figure 4 a), b) and figure caption, lines 4, 6, and 16, and Figure 7. The use of Celsius instead of K does not seem necessary in any of these cases, except perhaps in Tables 1 and 2, which give manufacturer specifications and may be given to whole Celsius values, but not the apparent 0.01 K precision that might result from calculation. I leave this to the authors to consider, but recommend that the choice is not made "by accident" in each individual case.

Reply: I can assure the referee that the choice between °C and K is not made "by accident". Of course, temperatures stated in Kelvin or degrees Celsius are equivalent. However, among metrology institutes it is common practice to use °C for absolute temperature values when the traceability to the International Temperature Scale of 1990 (ITS-90) shall be indicated and to use K and mK when (small) differences between temperatures or uncertainties shall be described. The Kelvin scale on Fig. 6 will be changed.

Comment:

p. 2 l. 29/30. Given the field of view of about 4 degrees x 4 degrees, it seems impossible not to include stars, galaxies or other in the "deep space" calibration point, which might degrade the uniformity of the temperature field. Please add a comment of explanation.

Reply: This aspect is very important and will be considered by the instrument developer. It is beyond the scope of this paper which only describes the large-area blackbody for the inflight calibration.

typos: p. 2 l. 14 consider "as an imaging FTS..." p. 2 l. 16 consider "... the chemistry mode, respectively."

Typos have been corrected.

---

## Author Response (AR1)

Comments on "A large-area blackbody for inflight calibration of an infrared interferometer deployed on board a long-duration balloon for stratospheric research"

Anonymous Referee #1

This paper describes a newly developed large-area blackbody which is planned to be equipped on a stratospheric balloon instrument. Although the specification and lab-test of the blackbody is well described, the current manuscript looks like too much technical, and it seems the content could be of interest to a rather limited range of readers. To increase the value of the paper for a wider area of the scientific community, I would recommend to add some more descriptions with respect to the following points:
(1) Please give a brief background-history about what kind of calibrators have been used in past similar instruments (for example, MIPAS Balloon)

- A brief background-history has been added as follows: "It is common practice for balloon-borne FTS instruments to establish onboard two-point calibration procedures using deep space as one reference point and a temperature stabilized blackbody {te02, friedl-vallon04}. That approach ensures the consistency of the measurements over a long period of time in a variable environment with changing detector sensitivity due to temperature alteration of the instrument."

(2) Adding an introduction about the science objectives of balloon-borne GLORIA may help the readers to understand why careful radiance calibration is required. I understand that this paper does not aim at discussing scientific aspects of the GLORIA mission, but still I consider that any instrumental designs are optimized based on the scientific requirements and such information is important for readers. I could not follow well from where all the instrumental requirements' numbers (those summarized in Table 1) come from.

- The science objectives of balloon-borne GLORIA instrument are similar to those of the airborne instrument. In the section "The GLORIA instrument" references to the scientific objectives and to the requirements for the calibration sources are given. The instrumental design of the balloon-borne GLORIA instrument has not been finalized yet.

- Minor comments
p.1 L18: ...the German metrology institute"," the spectral and ...

- Typo has been corrected

p.4 Figure 2: if possible, please add the scale (length) on the close-up figure of pyramids.

- The size of the pyramids has been added: Base is 5mm x 5mm; height = 9mm.

p.7 Figure 5(a): please add some labels...it's hard to understand what exists in the experimental layout.

- Figure has been modified

p.8 Figure 6: The vertical axis of figure may be expressed in Celsius degree so that the comparison with Figures 4 and 7 become more easy.

- Figure has been modified

Anonymous Referee #2

The manuscript describes the technical solution for calibrating a balloon-borne Fourier Transfom Spectrometer during its mission, including laboratory testing of the calibration hardware and procedure. The manuscript is well written, and the description meets scientific and academic standards, being understandable, repeatable, and with sufficient quantitative detail. I do agree with the very reasonable comments by the anonymous reviewer #1 and by Anne Kleinert. I recommend that the authors implement these in the final version and subsequently, I recommend the manuscript's acceptance.

I urge the authors to make a conscious decision on whether temperatures should be given in K, in degrees Celsius, or both. They may consider using K throughout, which would presumably be the most systematic solution. Perhaps it is worthwhile to add the Celsius value in parentheses in certain cases. Celsius (Centigrade) are now used on p. 4 in Table 1, p. 5 line 2 and Table 2, p. 6 Figure 4 a), b) and figure caption, lines 4, 6, and 16, and Figure 7. The use of Celsius instead of K does not seem necessary in any of these cases, except perhaps in Tables 1 and 2, which give manufacturer specifications and may be given to whole Celsius values, but not the apparent 0.01 K precision that might result from calculation. I leave this to the authors to consider, but recommend that the choice is not made "by accident" in each individual case.

- I can assure the referee that the choice between °C and K is not made "by accident". Of course, temperatures stated in Kelvin or degrees Celsius are equivalent. However, among metrology institutes it is common practice to use °C for absolute temperature values when the traceability to the International Temperature Scale of 1990 (ITS-90) shall be indicated and to use K and mK when (small) differences between temperatures or uncertainties shall be described. The Kelvin scale on Fig. 6 has been changed.

p. 2 l. 29/30. Given the field of view of about 4 degrees x 4 degrees, it seems impossible not to include stars, galaxies or other in the "deep space" calibration point, which might degrade the uniformity of the temperature field. Please add a comment of explanation.

- This aspect is very important and will be considered by the instrument developer. It is beyond the scope of this paper which only describes the large-area blackbody for the inflight calibration.

typos:

p. 2 l. 14 consider "as an imaging FTS..."

p. 2 l. 16 consider "... the chemistry mode, respectively."

- Typos have been corrected

Interactive comment by A. Kleinert

This paper presents a large-area blackbody designed for application on a stratospheric balloon. It is temperature stabilized using a phase change material. Some more explanations on the characterization measurements and their results would increase the value of the paper for the scientific community:

P.1 l.18: "... the spectral and spatial radiance distribution of the blackbody was determined."
Can you give some results of the spectral distribution?
- ■ The word "spectral" has been deleted

P.4 Tab.1: Here the required spatial temperature uniformity is < 0.15 K, while elsewhere in the text the requirement is given as < 0.1 K.
- ■ It has been corrected

P.6 Fig.4: In the left plot, the plateau is at -30 °C, while it is at -32 °C in the right plot.
What is the reason for this difference?
- ■ These measurements were made in very different environments. At ambient pressure in the lab convection has a large effect on the temperature. The important issue is the long-term stability not the absolute temperature.

P.6, l.15/16: "Due to its large optical surface, the blackbody faces a thermally non-uniform environment inside the RBCF with temperatures in the range between -120 °C and 23 °C."
Can you explain where these large temperature differences come from? Generally some more words on the RBCF and the measurement setup would be helpful.
- ■ The reason for the thermally non-uniform environment is the setup inside the RBCF. The cold aperture in front of the blackbody could not suppress all radiation coming from the walls.
  The following has been added to the text: "The cooled optical pathway suppressing the background radiation from the wall and surrounding the field of view of the VIRST along its line of sight is a sequence of cooled apertures and tubes. This aperture system (beamline) extends into the source chamber and has there an outer diameter of 40 mm. Because the opening of the balloon blackbody is 100 mm by 100 mm a part of this aperture "sees" the cooled beamline (-120 °C) and the remain aperture of the blackbody the inner walls of the source chamber which are at room temperature."

P.7, Fig.5a: Can you add some labels to the figure? Where is the source chamber, where is VIRST? And how is the temperature distribution inside the RCBF (see also comment above)?
- ■ Labels have been added to the figure and T-distribution is described in more detail in the text (see above)

P.7 l.2: "The collinearity of radiance temperature and contact temperature is obvious."
This is true, but there is a systematic difference of about 50 mK. Is this expected?
- ■ Yes. This systematic difference is due to the fact that there is a thermal gradient between the aluminium base and coated optical surface.
  The following has been added to the text: "The slightly higher radiance temperature results from the thermal gradient between sensor and emitting surface due to the thermal conductivities of the aluminium and the coating and the non-ideal emissivity of the BBB. Due to the later a part of the radiation from the surrounding (1 – emissivity) contributes to the apparent radiance temperature and leads to higher apparent temperatures when the BBB is operated below room temperature."

P.7 l.12: "the corresponding radiance temperature is given as well by the calibration in terms of radiance temperature"

What is meant by "the calibration in terms of radiance temperature"?
- Has been changed to: "characterization with VIRST."

P.8 Fig.6: At the beginning of the measurement, the temperature of the external chiller is just around the freezing temperature of the PCM. Are you sure that the PCM is completely frozen?
- Yes. The external chiller has been operated at a lower temperature before.

As soon as the chiller temperature is raised, the temperature of the blackbody rises rather fast for about half an hour before reaching the melting plateau. I would expect the melting to start immediately. Can you comment on this behaviour?
- That phase is not relevant for the calibration. But this is the explanation: Due to the thermal conductivity (thermal resistance) between the heat-exchanger of the chiller and the inner volume of the PCM elements the melting process starts delayed. During that delay no heat for melting is required and consequently the temperature in the aluminium and of the emitting surface rises faster.

The temperature gradient in this figure is about 75 mK/h which is more than twice as strong as in the lab (Fig. 4a) although the pressure is much lower and the ambient temperature is the same or lower. Is this expected?
- During the lab test the blackbody was completely closed in order to avoid condensation. There was no external radiation from the environment to the optical surface. So those are completely different setups.

P.9 Fig.7: The figure shows only 70 mm x 70 mm of the 125 mm x 125 mm optical surface. How is the temperature distribution outside the range shown?
- The temperature is higher in the outer areas not shown in the figure. But this could be attributed to a vignetting of the optical path of the VIRST and is therefore not representative for the true lateral temperature distribution of the backplane. Therefore this data is not shown here.

Is the systematic gradient between upper and lower part of the optical surface expected? Is this gradient also reflected in the data of the 10 PRTs which are mentioned on p.3 l.11?
- During the calibration measurements there were numerous heat transfers in the system: coolant to the aluminium of heat exchangers, aluminium of heat exchangers to the plastic container of the PCM, from the plastic container to the PCM itself, from the PCM to the baseplate of the optical surface, baseplate to black coating. For sure, all those heat transfers are not homogenous. The coolant has a lower temperature when it enters the heat exchanger than when it exit the system. It could also be the influence of the BBB walls which were cooled by a separate system. Furthermore there should a convective flow which should lead to a vertical gradient very similar to the one observed here.

During the measurement of the lateral distribution (gap in Fig. 6), the overall blackbody temperature rises by about 100 to 150 mK. How is this temperature rise taken into account in order to make sure that the distribution shown in Fig. 7 is really a lateral effect and not interfering with an effect in time?
- It took about two hours to measure the spatial distribution. The time effect could not be avoided. So the results reflect a worst-case scenario. (Since the aperture is scanned up-down for every column and then column-wise from left to right a possible drift should appear as horizontal gradient.)

Typos:
P.1 l.18: A comma is missing after "institute"

P.3, l.1: "influence" without "s

- ■ Typos have been corrected